# Overview of Trop-2 in Cancer: From Pre-Clinical Studies to Future Directions in Clinical Settings

**DOI:** 10.3390/cancers15061744

**Published:** 2023-03-13

**Authors:** Pasquale Lombardi, Marco Filetti, Rosa Falcone, Valeria Altamura, Francesco Paroni Sterbini, Emilio Bria, Alessandra Fabi, Diana Giannarelli, Giovanni Scambia, Gennaro Daniele

**Affiliations:** 1Phase 1 Unit, Fondazione Policlinico Universitario A. Gemelli IRCCS, 00168 Rome, Italy; 2Department of Experimental Medicine, Sapienza University of Rome, 00185 Rome, Italy; 3Comprehensive Cancer Center, Fondazione Policlinico Universitario A. Gemelli IRCCS, 00168 Rome, Italy; 4Department of Translational Medicine and Surgery, Universitá Cattolica del Sacro Cuore, 00168 Rome, Italy; 5Precision Medicine in Senology, Fondazione Policlinico Universitario A. Gemelli IRCCS, 00168 Rome, Italy; 6Facility of Epidemiology and Biostatistics, Fondazione Policlinico Universitario A. Gemelli IRCCS, 00168 Rome, Italy; 7Department of Life Science and Public Health, Università Cattolica del Sacro Cuore, 00168 Rome, Italy; 8Scientific Directorate, Fondazione Policlinico Universitario A. Gemelli IRCCS, 00168 Rome, Italy

**Keywords:** Trop-2, ADC, target therapy, bystander effect

## Abstract

**Simple Summary:**

Trophoblast cell surface antigen-2 (Trop-2) is a widely expressed glycoprotein on a variety of different tumours. Trop-2 is considered as a marker of germ cells and is associated with regenerative ability in several tissues. Some studies demonstrated both oncogenic and tumour suppressor roles for Trop-2. In recent years, the therapeutic value of Trop-2 was identified and various studies with drug–antibody conjugates have been pursued in cancer patients. In this work, we reviewed both the pre-clinical and clinical activities of anti-Trop-2 therapy to highlight the future developments of these therapies.

**Abstract:**

Trophoblast cell surface antigen-2 (Trop-2) is a glycoprotein that was first described as a membrane marker of trophoblast cells and was associated with regenerative abilities. Trop-2 overexpression was also described in several tumour types. Nevertheless, the therapeutic potential of Trop-2 was widely recognized and clinical studies with drug–antibody conjugates have been initiated in various cancer types. Recently, these efforts have been rewarded with the approval of sacituzumab govitecan from both the Food and Drug Administration (FDA) and European Medicines Agency (EMA), for metastatic triple-negative breast cancer patients. In our work, we briefly summarize the various characteristics of cancer cells overexpressing Trop-2, the pre-clinical activities of specific inhibitors, and the role of anti-Trop-2 therapy in current clinical practice. We also review the ongoing clinical trials to provide a snapshot of the future developments of these therapies.

## 1. Background

### 1.1. TROP-2 Biology and Functions

The transmembrane glycoprotein trophoblast cell surface antigen-2 (Trop-2) is widely expressed in various epithelial cancers as well as in specific normal tissue. Trop-2 is also known as tumour-associated calcium signal transducer 2 (TACSTD2), membrane component chromosome 1 surface marker 1 (M1S1), gastrointestinal antigen 733-1 (GA733-1), and epithelial glycoprotein-1 (EGP-1) [1].

Trop-2 was initially discovered in placental trophoblastic tissue, and the cells expressing this biomarker have the capacity to invade the uterus during placental implantation [2,3]. Lipinski et al. identified four new transmembrane glycoproteins (Trop-1, 2, 3, and 4) expressed on normal and malignant embryonal cells and, among them, only Trop-2 may similarly confer the capacity for proliferation and invasion to cancer cells [4].

Although the physiological function of Trop-2 is not fully clarified and is still under investigation, Trop-2 is implicated in several intracellular axes, including the MAPK/PI3K/AKT pathways that are implicated in proliferation, migration, and invasion of cancer cells [5,6,7]. Overexpression of Trop-2 was associated with accelerated tumour growth and a dismal prognosis in various types of cancers, including breast, gastric, and ovarian cancers [8,9,10]. Conversely, in other tumours like non-small cell lung cancer (NSCLC), TROP-2 downregulation and internalization into the cytoplasm, are related to metastasis and recurrence [11]. Trop-2 is also overexpressed in haematologic malignancies like as leukaemia, extranodal nasal type lymphoma (ENK/TL), and non-Hodgkin’s lymphoma (NHL) [12,13].

These characteristics could make Trop-2 a seductive target for cancer therapy. Currently, numerous therapeutic strategies with antibodies or antibody–drug conjugates are being developed to target Trop-2 in specific tumours.

### 1.2. Trop-2 Properties, Binding Partners, and Signalling Pathways

Trop-2/TACSTD2 was first described in 1981 and its gene is located on chromosome 1p32 [2,4]. The tertiary structure of Trop-2 consists of multiple domains that extend through the cell membrane. The extracellular domain is composed of a 26-amino acid hydrophobic peptide and an N-terminal part, the largest part of the molecule consisting of 274 amino-acids, also known as the ectodomain (Trop-2EC). It is comprised of an epidermal growth factor-like repeat containing a cysteine-rich domain, a thyroglobulin type-1 domain, and a cysteine-poor domain, anchored via a single transmembrane helix (TM) followed by a short intracellular tail (Trop-2IC) [14].

The 30-amino acid cytoplasmic part shows high homology to a HIKE domain [15,16] and includes a serine residue (S303) that is phosphorylated by protein kinase C (PKC) [17] and a phosphatidyl-inositol 4,5-bisphosphate (PIP2) binding site [8].

Trop-2 is a member of a protein family (GA733 family) that includes at least two “type I” membrane proteins: GA733-1 (Trop-2) and GA733-2, also known as EpCAM (epithelial cell adhesion molecule). Trop-2 and EpCAM exhibit very high similarities in sequence and structure, with 49% homology and 65% similarity in amino acid repeats and a comparable arrangement of hydrophilic and hydrophobic parts [14,18]. Nevertheless, the promotor regions of EpCAM and Trop-2 are unrelated, resulting in different expression patterns [19] and leading to structural differences in the intracellular domain explaining the distinct intracellular signalling and functions between Trop-2 and EpCAM [20,21]. Indeed, EpCAM exhibits its role in cell differentiation, proliferation, and migration through c-myc. On the contrary, Trop-2 has been reported to interact with several proteins, such as insulin-like growth factor-1 (IGF-1) 11, claudin-1 and 7, cyclin D1, and PKC. Furthermore, due to the HIKE domain, the PIP2 binding site, and the serine phosphorylated by PKC, Trop-2 is involved in calcium signalling through which the MAPK pathway could be activated [5].

#### 1.2.1. IGF-1/IGF-1R

IGF-1, as mentioned above, binds Trop-2 leading to modulation of IGF-1 signalling and activation through PIP2 and Ca^2+^. Trop-2 may also bind the receptor of IGR-1 (IGF-1R), blocking IGF-1 signalling [11] and playing critical roles in cell growth, differentiation, transformation, and metastasis. This mechanism could explain the different impacts of Trop-2 overexpression in lung cancer. Indeed, in a lung cancer model, high expression of Trop-2 suppressed tumour growth by attenuating IGF-1R signalling, likely by binding IGF-1 [14].

#### 1.2.2. Claudin

Claudin-1 and 7, two transmembrane proteins composing the tight junctions at the epithelial surface, bind to Trop-2′s ectodomain preventing claudin degradation which plays a fundamental role in epithelial barrier maintenance. Trop-2 might also indirectly affect adhesive interactions between cells by modulating the complex formation between fibronectin and P1 integrin/RACK1 (receptor for activated PKC) [14].

#### 1.2.3. ERK1/2

Trop-2 could also initiate the ERK1/2-MAPK axis, leading to malignant transformation [11], and could dysregulate stem cell function via the Notch, Hedgehog, and Wnt pathways through the expression and activation of cyclic AMP-responsive element-binding protein (CREB1), Jun, NF-κB, Rb, STAT1, and STAT3 (Figure 1) [5]. As previously mentioned, the MAPK pathway is stimulated by increased Ca^2+^ and phosphorylation of MAPK, which affects cell cycle progression. Furthermore, ERK activation was observed in various tumour types characterized by Trop-2 overexpression, and this ERK1/2 activation is thought to promote tumour survival through anti-apoptotic effects [5].

#### 1.2.4. NRG1

Trop-2 seems to also directly affect NRG1 via the EGF-like and thyroglobulin repeat domains in the extracellular region of Trop-2. In a pre-clinical model of HNSCC cells, Zhang and colleagues knocked down Trop-2 resulting in increased membrane localization of NRG1 that binds to and increases the activation of ErbB3 [22].

#### 1.2.5. Cyclin D1

Trop-2 could create a fusion product with cyclin D1 (bicistronic cyclin D1/Trop-2) [23] due to post transcriptional processes. This change could affect the stability of cyclin D1 increasing cell longevity and proliferation and qualifying Trop-2 as an oncogene.

#### 1.2.6. PKC

Lastly, PKC phosphorylates Trop-2′s cytoplasmic tail at S303. PIP2 has been suggested as a phosphorylation regulator of S303 by PKC through activation of the Raf and NF-κB pathways, promoting cancer cell survival [24].

## 2. TROP-2 Significance in Cancer

Despite Trop-2 being first identified as a cell surface marker for trophoblast cells, a great effort has been made to elucidate the role of this marker in cancers [21].

Trop-2 was also detected in stem cells of different tissues, especially in basal cells. As an example, the expression of TROP- 2 had been associated with self-renewal, regeneration, and differentiation properties in prostate basal cells [25,26], oval cells after liver injury [27], and endometrium-regenerating cells [28]. These findings support the role of Trop-2 as a regulator of stem cell growth which is implicated in the regeneration of various tissues, perhaps playing a role in physiological events like hyperplasia.

On the other hand, as noted above with few exceptions, its overexpression has also been associated with the increase in tumour growth, proliferation, and metastasis in various epithelial cancers, i.e., head and neck, thyroid, lung, gastrointestinal tract, breast, renal, and gynaecological cancers, and glioma [29].

Beyond the prognostic value, the real weight of Trop-2 gain of function or Trop-2 loss in tumorigenesis, epithelial–mesenchymal transition (EMT), and mesenchymal trans-differentiation remains unclear [30,31].

Silencing the Trop-2 gene using small interfering (si) RNA in colon, breast, cervical, lung, and ovarian cancer cell models led to a suppression of malignant transformation inhibiting the proliferation, invasion, and the formation of colonies in vitro [30,32,33,34,35]. The knockdown of Trop-2 in gallbladder cancer inhibited vimentin and increased E-cadherin expression linked to EMT [36]. Trop-2 overexpression seems to be related to an increased risk of metastasis in patients affected by various cancer types (oral squamous, thyroid, some oesophageal, gastric, colorectal, pancreatic, ovarian, uterine, cervical, prostate, and urinary bladder). However, it is not upregulated in others (e.g., head and neck and certain lung cancers, such as lung adenosquamous and squamous cell carcinoma) [1,29,37,38,39].

In gastric cancer, the poor prognostic value seems to be related to the co-expression of Trop-2 and amphiregulin [40], while tumour necrosis factor-α (TNF-α) axes could regulate this effect in colorectal cancer. Indeed, a low concentration of this cytokine correlates with an increase in Trop-2 protein expression, while higher concentrations of TNF-α reduce migration and cell invasiveness [41]. These authors linked these activities to the crosstalk between TNF-α and the ERK1/2 pathway, showing that an ERK1/2 inhibitor can suppress the cytokine’s upregulation of Trop-2 [41]. The ERK1/2 pathway was also involved in pancreatic cancer, gynaecological cancers, and HNSCC, where Trop-2 expression increases the phosphorylation of ERK1/2 leading to the activation of the ERK/MAPK pathway, increasing the levels of cyclin D1 and cyclin E that resulted in a cell cycle dysregulation [5,32,42]. High cyclin D1 expression seems to also be the result of the activation of Trop-2 via in breast cancer [32,43].

In a prostatic model, Goldstein et al. showed that a malignant transformation could arise from the Trop-2+ basal cells in immunodeficient mice [26]. Thus, basal cells expressing Trop-2 and CD44 can develop luminal phenotype tumours [25,26,44,45]. This is consistent with studies implicating Trop-2 as a critical regulator of β1 integrin activities and promoting prostate cancer cell motility [3,46]. Interestingly, Trop-2+ exosomes purified from prostate cancer promote migration of Trop-2-negative prostate cancer cells on fibronectin, suggesting that Trop-2 could induce cells lacking Trop-2 to gain Trop-2 regulatory properties affecting migration [47].

The role of Trop-2 in haematological disease is still unclear, since it is expressed in Hodgkin’s lymphoma and chronic lymphocytic leukaemia^1^, but not in anaplastic large cell lymphoma [13].

To summarize, the Trop-2 gene is related to several transcription factors leading to a dysregulation of the numerous pathways connected with this glycoprotein. Although not yet fully elucidated, it activates CREB1, Jun, NF-κB, Rb, STAT1, and STAT3 via induction of the cyclin D1 and ERK/MEK pathways affecting malignant transformation and metastasis [8,33,48,49].

## 3. Strategies to Target Trop-2

Due to the ubiquitous expression in cancer cells with a relatively low expression in most normal tissues, Trop-2 represents an excellent candidate as diagnostic [50,51,52] and a therapeutic target, specifically for antibody-based therapy. Nevertheless, relevant toxicities have been reported relating to the role of Trop-2 in healthy tissues [37].

Its overexpression in advanced tumours represents an attractive and promising therapeutic target [24]. Moreover, Trop-2 is highly expressed in primary neoplasms and has been proposed to be a landmark for undifferentiated epithelial cells [48]. Several therapeutics have been designed considering TROP-2 as a target. Although the most advanced in the clinical setting is represented by antibody–drug-conjugates (ADCs) (Table 1 and Table 2).

### 3.1. Sacituzumab Govitecan

Sacituzumab govitecan is an ADC composed of an anti-Trop-2 humanized monoclonal antibody hRS7 IgG1k coupled with SN-38, an active metabolite of irinotecan (IRI), a topoisomerase I inhibitor.

This ADC is characterized by a high drug-to-antibody ratio (DAR) (7.5–8 molecules of SN-38 conjugated to each antibody via the unique hydrolysable and proprietary linker, CL2A), permitted by the significantly better toxicity profile of SN-38 compared with the parent drug irinotecan. Furthermore, this ADC also provides an extracellular release of the drug (payload) that kills surrounding or bystander cells, which do not necessarily express Trop-2 (i.e., bystander killing effect) [53].

In cell models, sacituzumab govitecan has demonstrated promising results against human epithelial cancer xenograft models of several cancers including gastrointestinal cancers (i.e., gastric and pancreatic cancers), NSCLC, breast cancer (BC), prostate, and cervical cancers [54]. Compared with either SN-38, IRI, or a non-targeting ADC, sacituzumab govitecan provided outstanding results in nearly every model studied [55,56,57].

Exceptions are represented by the SK-MES-1 cell line (derived from squamous cell carcinoma) and MDA-MB-231 [derived from triple negative breast cancer (TNBC)] cell line. In these models, the anti-proliferative effect of sacituzumab govitecan was low and not significantly different from IRI. These cell lines both have low expression of Trop-2 on their cell membranes. Indeed, in order to verify this evidence, MDA-MB-231 cell lines were engineered to express high levels of Trop-2.

Interestingly, the transfection of the MDA-MB-231 cells with human Trop-2 cDNA did not appreciably modify their sensitivity to IRI, but increased Trop-2 expression ~4 fold (from ~30,000 to 120,000), enhancing sacituzumab govitecan’s therapeutic activity [58]. The high specificity against Trop-2 of this ADC drove this enhancement since responses to a non-targeting ADC were not significantly different between the engineered and non-engineered cells. Another study highlighted an increased delivery of SN-38 via the anti-Trop-2 antibody, compared to IRI. Sharkey et al. showed that the levels of SN-38 in tumour tissues were between 20- and 136-fold higher for the sacituzumab govitecan compared to IRI. These data show a substantial advantage in using an ADC strategy to release the topoisomerase-1-inhibiting drug [59].

An important advantage of the ADC therapeutic strategy is to reduce collateral damage to normal tissues due to the high DAR. Adverse events represent an essential topic due to Trop-2′s expression in normal tissues. The safety studies were efficiently performed in Cynomolgus monkeys thanks to a similar Trop-2 expression level as humans. At the highest dosage of sacituzumab govitecan tested, the primates experienced severe but reversible neutropenia and diarrhoea, symptoms compatible with SN-38/IRI toxicity. Moreover, a study conducted by Sharkey et al. demonstrated that the levels of glucuronidated SN-38 (SN-38G), a detoxified derivative of SN-38, in the animals’ serum was much lower with the ADC. The lower levels of SN-38G found in serum suggested severe adverse events could have been reduced with sacituzumab govitecan with respect to IRI [60].

### 3.2. Datopotamab Deruxtecan (DS-1062a)

Datopotamab is a human IgG1 mAb Trop-2-directed ADC with a potent Topo I inhibitor, deruxtecan (DXd), conjugated through a tetrapeptide-based linker. This ADC has a DAR of 4 that is expected to maximize the therapeutic window. The tetrapeptide-based linker is enzymatically cleavable and designed to release DXd after proteolytic processing by lysosomal enzymes such as cathepsins [61].

In vitro, Dato-DXd inhibited cell growth in several cancer cell lines from multiple tumour types (pharynx, pancreas, ovary, stomach, breast, and lung). Dato-DXd markedly inhibited the cell proliferation of Trop-2-high cell lines but was ineffective against Trop-2-low cell lines. These results were confirmed in a Trop-2-positive xenograft mouse model with a single dose of Dato-DXd at 10 mg/kg that significantly inhibited tumour growth inhibition by 96%. On the contrary, neither datopotamab nor control ADC inhibited tumour growth at the same dose. In fact, in tumour cell models treated with Dato-DXd, an increased level of H2AX and KAP1 phosphorylation, which are DNA damage markers, were observed in a time-dependent manner until day 7. This phenomenon was not observed in tumours treated with control ADC or datopotamab.

The safety profile of Dato-DXd was evaluated in murine and primate models. Pulmonary toxicity was observed at ≥30 mg/kg in monkey models and was characterized by cell infiltration, oedema, and fibrosis. Furthermore, Dato-DXd induced only slight intestinal or hematopoietic toxicity in rats and monkeys and without severe changes even at the maximum feasible doses, likely due to reduced off-target toxicity by the stable linker adopted for Dato-DXd. Nevertheless, hyperpigmentation in the epidermis (not reversible) and corneal lesions including single-cell necrosis and pigmentation in the epithelium (reversible) were observed in the skin and cornea at ≥30 mg/kg in monkeys [61].

### 3.3. PF-06664178

PF-06664178 is composed of a humanized IgG1 antibody conjugated to microtubule inhibitor AcLys-VCAur0101 linker-payload at the C-terminus of the antibody heavy chain. Upon binding to Trop-2 in the extracellular portion on the cell surface, PF-06664178 is internalized to the lysosomes and processed by proteases to release its auristatin-based Aur0101 payload [62,63].

In vivo studies showed significant antitumor drug activity and an IC50 below 1 nmol/L for most cell lines. Like other ADCs, even PF-06664178 demonstrated a low activity against Trop-2-negative cell lines, suggesting that Trop-2 expression is required for the ADC’s antitumor activity.

In vivo studies on patient-derived tumour xenograft models of pancreatic, lung, ovarian, and triple-negative breast cancers demonstrate that doses showed a sustained regression ranging from 0.75 to 3 mg/kg with a single 1.5 mg/kg dose of PF-06664178 resulting in an area under the curve (AUC) of 0–336 percentage of ADC from the total antibody population in serum of 87%. This displays good payload stability in vivo, minimizing the side effects and avoiding the release of the payload in a non-target tissue. Consistent with in vitro studies, PF-06664178 was not effective against Trop-2-negative tumours.

Preliminary toxicology reports in monkeys (up to 6 mg/kg doses), demonstrated toxicity signals in non-tumoral tissues that express Trop-2 antigens. Reversible toxicities included necrosis in skin, upper gastrointestinal mucosa, and vagina epithelium [64].

### 3.4. SKB264

SKB264 has a humanized IgG1 mAb hRS7 conjugated with a proprietary cytotoxic, belotecan-derived payload and novel stable conjugation chemistry to achieve an average DAR of 7.4. The release of payload upon SKB264 internalization is proportional to the Trop-2 expression. The payload-linker is conjugated to cysteine residues, and the release of the payload relies on hydrolysis, which would happen in the tumour microenvironment, lysosome, and plasma [65].

In in vitro models, SKB264 showed significant anti-tumour activity in a dose-dependent manner with a significant tumour inhibitory activity at 1, 3, and 10 mg/kg. The efficacy of SKB264 seems to be higher than that of sacituzumab govitecan at doses of 1 and 3 mg/kg.

## 4. Clinical Experience

We searched The PubMed, Cochrane Library, and Embase databases to identify reports of trials with anti-TROP-2 drugs published up until October 2022.

We identified twelve papers reporting results from six trials (Table 1). All selected trials were prospective; most were non-randomized early phase studies (I-II) (4/6, 67%), while only two were randomized phase III trials. Ten of the papers reported data on sacituzumab govitecan.

Most publications concern a single phase I/II basket trial, the IMMU-132-01 trial [66], a single-arm, open-label, multi-centre phase I/II trial that involved dose-escalation, and a cohort expansion phase that enrolled patients with advanced disease. The primary endpoint was safety and pharmacokinetics, with an investigator-evaluated objective response rate (ORR) as a secondary endpoint.

A total of 495 patients were treated. The most represented histologies were TNBC (n = 144), HR+ BC (n = 68), small cell lung cancer (SCLC) (n = 62), NSCLC (n = 54), urothelial cancer (UC) (n = 49), and colorectal cancer (CRC) (n = 31). Overall, the treatment was fairly well-tolerated; 192 (38.8%) patients had serious AEs (SAEs) and in 75 patients (15.2%) SAEs were related to treatment. Most common treatment-related SAEs were febrile neutropenia (4.0%), diarrhoea (2.8%), and vomiting (1.4%).

The results of the metastatic BC [67,68], NSCLC [69], and SCLC [70] cohorts were published individually in separate publications. The NSCLC cohort showed an ORR of 19% and a clinical benefit rate of 43%, with more than half of the patients having been heavily pre-treated (at least three lines of treatment). The SCLC cohort demonstrated equally interesting activity signals, with an ORR of 14% and a clinical benefit rate (i.e., the percentage of patients with an objective response or stable disease ≥4 months) of 34% in the population of pre-treated patients. As compared with the standard topotecan yielding a modest 5% in patients progressing/relapsing <90 days from the previous treatment and 17% in those progressing after 90 days, sacituzumab demonstrated an improved ORR of 11% and 20%, respectively [71].

The TNBC cohort showed the most compelling results, with an objective response rate (ORR) of 33.3% and a durable duration of response (DOR) (median 7.7 months). The subsequent randomized phase III study ASCENT [72] confirmed the outstanding results of the expansion cohort, recording an ORR of 35% and a progression-free survival (PFS) and overall survival (OS) statistically higher than the control arm (5.6 vs. 1.7 months and 12.1 vs. 6.7, respectively). Based on the results of this study, sacituzumab govitecan was approved by the Food and Drug Administration (FDA) and European Medicines Agency (EMA) for the treatment of metastatic TNBC who have only received at least two prior therapies for metastatic diseases. In addition, two first line studies (ASCENT-03 and ASCENT-04) investigating sacituzumab govitecan either in monotherapy or in combination with pembrolizumab, are ongoing.

Furthermore, the results of the phase III TROPiCS-02 study, evaluating sacituzumab govitecan in patients with HR+/HER2- metastatic BC pre-treated with endocrine therapy, CDK4/6 inhibitors, and at least two but not more than four lines of chemotherapy, was recently published. The study met its primary endpoint with a 34% reduction in risk of progression or death (hazard ratio, 0.66 [95% CI, 0.53 to 0.83; *p* = 0.0003]). The median PFS was 5.5 months (95% CI, 4.2 to 7.0) with sacituzumab govitecan and 4.0 months (95% CI, 3.1 to 4.4) with chemotherapy [73]. Recently, OS data were updated to the ESMO congress in which sacituzumab govitecan demonstrated a statistically significant and clinically meaningful improvement in survival [median 14.4 vs. 11.2 months; HR, 0.79 (0.65–0.96); *p* = 0.020] [74]. Interestingly, TROPiCS-02 showed that sacituzumab govitecan is effective in patients with a wide range of Trop-2 expression levels. PFS and OS benefit seems to be maintained in both high and low Trop-2expression subgroups. In detail, when the histochemical score (H-score, range = 0–300) was set to 100, the median PFS was 5.3 months with sacituzumab govitecan and 4.0 months with the treatment of the physician’s choice (HR = 0.77, 95% confidence interval [CI] = 0.54–1.09) for <100 subgroup and, for the ≥ 100 subgroup, these rates were 6.4 months and 4.1 months, respectively (HR = 0.60, 95% CI = 0.44–0.81) [75].

The data from the phase II TROPHY-U-01 trial [76] were recently published. In the published cohort, patients with metastatic UC, who progressed after prior platinum-based chemotherapy and checkpoint inhibitors, were treated with sacituzumab govitecan. A total of 113 patients were enrolled, of which about 80% were pre-treated with at least two lines of treatment. The ORR, the primary endpoint of the study, was 27% (95% CI, 19.5 to 36.6), which included a 5.4% complete response rate. Based on these results, the FDA granted accelerated approval to sacituzumab govitecan for metastatic UC patients after platinum-based chemotherapy and checkpoint inhibitors.

## 5. Future Directions

We searched ongoing clinical trials with anti-TROP-2 drugs on clinicaltrials.gov. Up to October 2022, we identified 48 studies (Table 2 and Table 3). Almost the totality of the studies (44/48, 91.7%) concerned the advanced/metastatic setting, with only four in neoadjuvant/adjuvant breast cancer.

The majority (52/66, 78.8%) of trials were early phase (I, I/II or II) studies. BC was the disease with the highest number of phase III trials. Randomization was present in 30.3% of trials. Anti-TROP-2 therapy was administered as monotherapy in one third of studies (36.4%). In the majority of the combination studies, immunotherapy represent the most common category.

Five anti-Trop-2ADCs are being investigated in clinical trials: sacituzumab govitecan, datopotamab deruxtecan (Dato-DXd, DS-1062a), JS108, SKB264, and STI-3258. All of them are administered intravenously. The first two ADCs are more widely used in the more advanced phases of development. In particular, the majority of the ongoing studies are investigating sacituzumab govitecan in different settings and/or diseases [77]. We could observe a substantial growth in the studies investigating this particular drug, increasing from 19 ongoing trials in October 2020 to 45 current studies [78].

The preliminary data of the two cohorts of the TROPION-Pan-Tumor01 study, a single-arm, multi-centre, phase I trial evaluating the safety and tolerability of datopotamab deruxtecan in pre-treated patients with epithelial tumours, were recently presented. The twenty-four patients affected by metastatic TNBC enrolled in the first cohort showed an ORR by BICR of 43% [79]. The disease control rate was 95%, with a manageable safety profile. In the second cohort, 133 patients affected by metastatic NSCLC were considered, of which 125 were evaluable. In total, an ORR of 36% was recorded with a DCR of 77%. Regarding the safety profile, 64 patients (48%) experienced grade 3 or higher TEAEs, and 12 patients (9%) had an interstitial lung disease judged to be treatment-related [80].

Encouraging results could be observed even in the metastatic NSCLC cohort of patients with actionable genomic alterations (i.e., EGFR, ALK, ROS1 and RET). Across the 34 heavily pre-treated patients, an ORR of 35% (95% CI, 19.7–53.5) was observed with a median DOR of 9.5 months (95% CI, 3.3–NE). Additionally, in this cohort the safety profile was manageable with nausea (62%) and stomatitis (56%) as common any-grade AEs [81].

## 6. Conclusions

In the past decade, Trop-2 was identified as a major regulator of multiple processes involved in carcinogenesis and tumour progression. Even though its role as an oncogene is currently debated, recent data suggest that Trop-2 acts both as a tumour promoter and tumour suppressor. The exact reason for these opposite observations remains to be determined.

Clinically, the therapeutic potential of Trop-2 is demonstrated by the fact that Trop-2 is overexpressed in most cancers, while healthy tissues express it only sporadically, making it an incredibly promising target for cancer-specific delivery of cytotoxic agents. Such a strategy is already utilized in several tumours, especially for those diseases where standard therapeutic options are limited (TNBC, UC) or ineffective (SCLC).

However, it should be noted that tumour heterogeneity could be a major limitation of the Trop-2-targeting therapeutic strategy. Further pre-clinical and clinical studies are necessary to clarify the fate and properties of this cell population in the response and resistance to Trop-2-targeting therapies. A valuable way to overcome these limitations is to administer anti-Trop-2 ADCs in combination with other drugs, with this optimal combination currently under investigation.

## Figures and Tables

**Figure 1 cancers-15-01744-f001:**
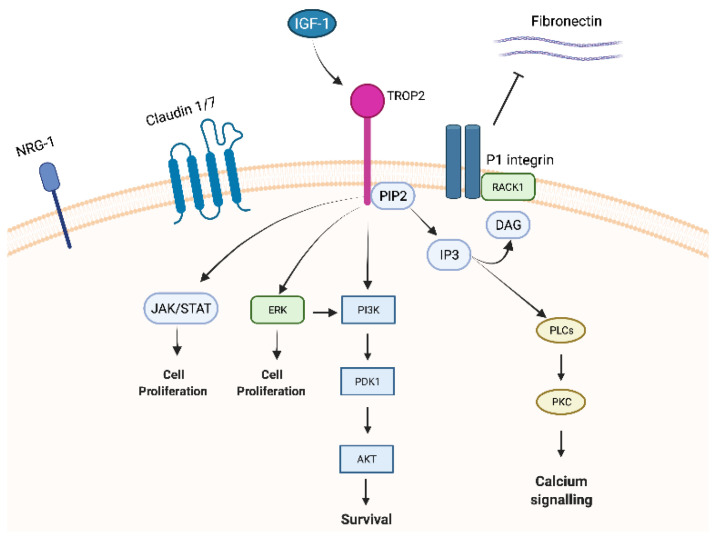
Trop-2 membrane-associated interacting partners and signalling pathways.

**Table 1 cancers-15-01744-t001:** Published clinical trials with anti-TROP2 drugs.

Clinical Trial (NCT)	Year of Publication	Phase	Randomization	Disease	Setting	Experimental Arm	Control Arm	Primary Objective	Population (n°)	Results	Safety—≥G3 Event (%)
IMMU-132-01 (NCT01631552)	2021	I/II	no	solid tumor	Advanced	Sacituzumab Govitecan	NA	AE/SAE, RDE, ORR, DOR, CBR, PFS, OS	495	ORR TNBC 33.3 (24.6–43.1); HR+ BC (31.5 (19.5–45.6); UR 28.9 (16.4–44.3); EC 22.2 (6.4–47.6); SCLC 17.7 (9.2–29.5); NSCLC 16.7 (7.9–29.3); CRPC 9.1 (0.2–41.3);	Neutropenia (42.4%), febrile neutropenia (5.2%), anemia (10.3%), diarrhea (7.9%)
ASCENT (NCT02574455)	2021	III	yes	TNBC	≥2 lines	Sacituzumab Govitecan	eribulin, vinorelbine, capecitabine, or gemcitabine	PFS	468 (235 and 233)	ORR 35% vs. 5%, experimental vs. control arm. PFS 5.6 vs. 1.7 months (hazard ratio for disease progression or death, 0.41; 95% CI, 0.32 to 0.52; *p* < 0.001). OS 12.1 vs. 6.7 months (hazard ratio for death, 0.48; 95% CI, 0.38 to 0.59; *p* < 0.001).	neutropenia (51% vs. 33%), leukopenia (10% vs. 5%), diarrhea (10% vs. <1%), anemia (8% vs. 5%), and febrile neutropenia (6% vs. 2%).
TROPiCS-02 (NCT03901339)	2022	III	yes	HR+/HER2- BC	Advanced	Sacituzumab Govitecan	eribulin, vinorelbine, capecitabine, or gemcitabine	PFS	543	PFS was 5.5 vs. 4.0 months HR, 0.66; (95% CI, 0.53 to 0.83; *p* = 0.0003)	neutropenia (51% vs. 38%), leukopenia (9% vs. 5%), diarrhea (9% vs. 11%), anemia (6% vs. 3%), and febrile neutropenia (5% vs. 4%).
TROPHY-U-01 (NCT03547973)	2021	II (Cohort I)	no	UC	≥2 lines	Sacituzumab Govitecan	NA	ORR per BICR	Cohort I: 113 patients	ORR 27%; mDoR 7.2 months (95% CI, 4.7 to 8.6 months) PFS and OS of 5.4 months (95% CI, 3.5 to 7.2 months) and 10.9 months (95% CI, 9.0 to 13.8 months), respectively.	neutropenia (35%), leukopenia (18%), anemia (14%), diarrhea (10%), and febrile neutropenia (10%), with 6% discontinuing treatment because of treatment-related adverse events.
NCT02122146	2018	I	no	solid tumor	Advanced	PF-06664178	NA	MTD, RP2D	31	ORR 0%	Neutropenia 19.3%; rash 12.9%
SEASTAR (NCT03992131)	2022	Ib	no	solid tumor	Advanced	Sacituzumab Govitecan plus Rucaparib	NA	MTD, RP2D	6	ORR 50% (PR 3/6)	Neutropenia (83.3.%); WBC count decrease (33.3%); Platelet count decrease (16.7%); Dyspnea (16.7%); Anemia (16.7%);

MTD: maximum tolerated dose; RP2D: recommended phase 2 dose; AE: adverse events; SAE: serious adverse events; RDE: recommended Doses for Expansion; ORR: objective response rate; Treatment-emergent Adverse Event (TEAE); DOR: duration of response; PR: partial response; CBR: clinical benefit rate; PFS: progression-free survival; OS: overall survival; TNBC: triple negative breast cancer; BC: breast cancer; HR+/HER2- BC: hormone Receptor–Positive/Human Epidermal Growth Factor Receptor 2–Negative Metastatic Breast Cancer; NSCLC: no small cell lung cancer; SCLC: small cell lung cancer; UC: urothelial cancer; HR: hazard ratio.

**Table 2 cancers-15-01744-t002:** Ongoing clinical trials with Sacituzumab Govitecan.

Clinical Trial Number	Phase	Randomization	Disease	Setting	Experimental Arm	Control Arm	Primary Objective	Status
NCT03995706	0/I	No	Breast Brain Metastasis and Glioblastoma	Advanced	Sacituzumab Govitecan	NA	PK	Recruiting
NCT05143229	I	No	Breast cancer	Advanced	Sacituzumab Govitecan plus Alpelisib	NA	RP2D, DLT	Not yet recruiting
NCT04724018	I	No	Urotelian Carcinoma	Advanced	Sacituzumab Govitecan Plus Enfortumab	NA	MTD, DLT	Recruiting
NCT04617522	I	No	Solid tumour and Moderate Liver Impairment	Advanced	Sacituzumab Govitecan	NA	TEAEs, PK, % of patientents with Clinically Significant Laboratory Abnormalities, Positive Anti-Sacituzumab Govitecan-hziy Antibodies	Recruiting
NCT05006794	Ia/b (Cohort of a Multi Arm Trial)	No	Triple-negative Breast Cancer, NSCLC	Advanced	Sacituzumab Govitecan plus GS-9716	NA	AEs, DLT	Recruiting
NCT04039230	I/II	No	Triple Negative Breast Cancer	Advanced	Sacituzumab Govitecan plus Talazoparib	NA	DLT	Recruiting
NCT04826341	I/II	No	Small Cell Lung Cancer and PARP inhibitor resistant tumors	Advanced	Sacituzumab Govitecan Plus Berzosertib	NA	ORR, MTD	Recruiting
NCT04927884	Ib/II	No	Triple Negative Breast Cancer	Advanced	Sacituzumab Govitecan plus cyclophosphamide, N-803, and PD-L1 t-haNK	NA	MTD, safety, ORR	Recruiting
NCT04863885	I/II	No	Urothelial Carcinoma	Advanced	Sacituzumab Govitecan plus Ipilimumab and Nivolumab	NA	MTD, ORR	Recruiting
NCT05101096	I/II	No	Solid Tumors or Triple-negative Breast Cancer (Japanese population)	Advanced	Sacituzumab Govitecan	NA	TEAEs, AEs, DLTs, ORR	Recruiting
NCT03992131	Ib/II	No	Solid Tumor	Advanced	Sacituzumab Govitecan plus Rucaparib	NA	AEs, DLT, ORR	Active, not recruiting
NCT03869190	Ib/II (Cohort of a Multi Arm Trial)	No	Urothelial Cancer	Advanced	Sacituzumab Govitecan plus Atezolizumab	NA	pCR, ORR	Recruiting
NCT03424005	Ib/II (Cohort of a Multi Arm Trial)	Yes	Triple-negative Breast Cancer	Advanced	Sacituzumab Govitecan plus Atezolizumab	NA	ORR, AEs	Recruiting
NCT03337698	Ib/II (Cohort of a Multi Arm Trial)	Yes	NSCLC	Advanced	Sacituzumab Govitecan plus Atezolizumab	NA	ORR	Recruiting
NCT04251416	II	No	Endometrial cancer	Advanced	Sacituzumab Govitecan	NA	ORR	Recruiting
NCT05119907	II	No	Oesophageal squamous-cell carcinoma, gastric and cervical adenocarcinoma	Advanced	Sacituzumab Govitecan	NA	ORR	Recruiting
NCT03964727	II	No	Solid tumor	Advanced	Sacituzumab Govitecan	NA	ORR	Recruiting
NCT04559230	II	No	Glioblastoma	Advanced	Sacituzumab Govitecan	NA	PFS	Recruiting
NCT04647916	II	No	Breast cancer with brain metastases	Advanced	Sacituzumab Govitecan	NA	ORR	Recruiting
NCT04230109	II	No	Localized Triple-Negative Breast Cancer	Localized	Sacituzumab Govitecan with or without Pembrolizumab	NA	pCR	Active, not recruiting
NCT04434040	II	No	Localized Triple Negative Breast Cancer	Localized	Sacituzumab Govitecan plus Atezolizumab	NA	Rate of undetectable circulating tumor cfDNA	Recruiting
NCT04448886	II	Yes	Hormone Receptor-positive (HR+)/HER2- Breast Cancer	Advanced	Sacituzumab Govitecan with or without Pembrolizumab	NA	PFS	Recruiting
NCT04468061	II	Yes	PD-L1 negative, triple Negative Breast Cancer	Advanced	Sacituzumab Govitecan With or Without Pembrolizumab	Sacituzumab Govitecan	PFS	Recruiting
NCT05113966	II	No	Triple Negative Breast Cancer	Advanced	Sacituzumab Govitecan plus trilaciclib	NA	PFS	Recruiting
NCT05520723	II	No	Triple Negative Breast Cancer	Advanced	Sacituzumab Govitecan plus loperamide	-	AE	Not yet recruiting
NCT04454437	IIb	No	Triple-negative Breast Cancer (Chinese population)	Advanced	Sacituzumab Govitecan	NA	ORR	Active, not recruiting
NCT03725761	II	No	Prostate cancer	Advanced	Sacituzumab Govitecan	NA	PSA response rate	Recruiting
NCT03547973	II	No	Urothelial Cancer	Advanced	Sacituzumab Govitecan with or without Pembrolizumab or Ciplatin plus Avelumab	NA	ORR	Recruiting
NCT05581589	II	No	Bladder Cancer	Localized	Sacituzumab Govitecan	-	pCR	Not yet recruiting
NCT05226117	II	No	Localized Bladder Cancer	Localized	Sacituzumab Govitecan	NA	pCR	Recruiting
NCT05535218	II	No	Bladder Cancer	Localized	Sacituzumab Govitecan plus Pembrolizumab	-	pCR	Not yet recruiting
NCT05186974	II	No	NSCLC Without Actionable Genomic Alteration	Advanced	Sacituzumab Govitecan with or without Pembrolizumab or Pembrolizumab plus Cisplatin or Carboplatin	NA	ORR; DLT	Recruiting
NCT03971409	Cohort of a Multi Arm Trial	No	Triple-negative Breast Cancer	Advanced	Sacituzumab Govitecan plus Avelumab	NA	Best Overall Response Rate (BORR)	Recruiting
NCT04958785	II (Cohort of a Multi Arm Trial)	No	Triple-negative Breast Cancer	Advanced	Magrolimab plus Sacituzumab govitecan	-	DLT. AEs. PFS, ORR	Recruiting
NCT05332561	II (Cohort of a Multi Arm Trial)	No	Triple-negative Breast Cancer	Localized	Sacituzumab Govitecan	-	IDFS	Not yet recruiting
NCT05327530	II (Cohort of a Multi Arm Trial)	Yes	Locally Advanced or metastatic Urothelial Cancer	Advanced	Sacituzumab Govitecan plus Avelumab	Avelumab	PFS, AEs	Recruiting
NCT05582499	Platform Series Study (Cohort of a Multi Arm Trial)	Yes	Breast Cancer	Localized	anti-TROP2 ADC NOS	Nab-paclitaxel plus Carboplatin	pCR	Not yet recruiting
NCT04527991	III	Yes	Unresectable Urothelial Cancer	Advanced	Sacituzumab Govitecan	Paclitaxel or Docetaxel or Vinflunine	OS	Recruiting
NCT05089734	III	Yes	NSCLC	Advanced	Sacituzumab Govitecan	Docetaxel	OS	Recruiting
NCT04595565	III	Yes	Localized Her2-negative, breast cancer	Localized	Sacituzumab Govitecan	capecitabine or platinum-based chemotherapy or observation	iDFS	Recruiting
NCT03901339	III	Yes	HR+, HER2-, Breast Cancer	Advanced	Sacituzumab Govitecan	Eribuline or Capecitabine or Gemcitabine or Vinorelbine	PFS	Active, not recruiting
NCT04639986	III	Yes	HR+, HER2-, Breast Cancer (Asian population)	Advanced	Sacituzumab Govitecan	Eribuline or Capecitabine or Gemcitabine or Vinorelbine	PFS	Recruiting
NCT05382286	III	Yes	Triple-negative Breast Cancer	Advanced	Sacituzumab Govitecan plus Pembrolizumab	Pembrolizumab plus Paclitaxel or Nab-paclitaxel or Gemcitabine	PFS	Recruiting
NCT05552001	IIIb	No	Triple-negative Breast Cancer	Advanced	Sacituzumab Govitecan	-	ORR	Not yet recruiting
NCT05382299	III	Yes	Triple-negative Breast Cancer	Advanced	Sacituzumab Govitecan	Paclitaxel or Nab-paclitaxel or Gemcitabine	PFS	Recruiting

MTD: maximum tolerated dose; RP2D: recommended phase 2 dose; DLT: dose limiting toxicity; AEs: adverse events; SAE: serious adverse events; RDE: recommended Doses for Expansion; ORR: objective response rate; TEAE: Treatment-emergent Adverse Event; iDFS: invasive disease free survival; NSCLC: no small cell lung cancer.

**Table 3 cancers-15-01744-t003:** Ongoing clinical trials with anti-TROP2 drugs other than Sacituzumab Govitecan.

Clinical Trial Number	Phase	Randomization	Disease	Setting	Experimental Arm	Control Arm	Primary Objective	Status
NCT05060276	I	No	Solid tumor	Advanced	STI-3258	NA	MTD, RP2D, DLT, AE/SAE	Not yet recruiting
NCT04152499	I-II	No	Solid tumor	Advanced	SKB264	NA	MTD, ORR	Recruiting
NCT05351788	II	No	NSCLC	Advanced	SKB264	-	AE, ORR	Not yet recruiting
NCT05445908	II	No	Triple-Negative Breast Cancer	Advanced	SKB264	-	ORR, AE	Not yet recruiting
NCT05347134	III	Yes	Triple-Negative Breast Cancer	Advanced	SKB264	Eribuline, or Capecitabine or Gemcitabine or Vinorelbine)	PFS	Not yet recruiting
NCT04601285	I	No	Solid tumor	Advanced	JS108	NA	DLT, MTD, AE	Recruiting
NCT03401385	I	Yes	Solid tumor	Advanced	Datopotamab Deruxtecan	NA	DLT, AE	Recruiting
NCT04612751	Ib	No	NSCLC	Advanced	Datopotamab Deruxtecan	NA	DLT, TEAE	Recruiting
NCT04526691	Ib	No	NSCLC	Advanced	Datopotamab Deruxtecan with Pembrolizumab With or Without Platinum Chemotherapy	NA	DLT, TEAE	Recruiting
NCT05460273	I/II	No	NSCLC, TNBC, Gastric cancer, Urothelial cancer and Other hystologies	Advanced	Datopotamab Deruxtecan	-	ORR	Recruiting
NCT04644068	I/IIa	No	Ovarian cancer	Advanced	Datopotamab Deruxtecan plus AZD5305	-	AE, DLT	Recruiting
NCT03742102	Ib/II (Cohort of a Multi Arm Trial)	No	Triple-Negative Breast Cancer	Advanced	Datopotamab Deruxtecan with Durvalumab	NS	AEs, ORR	Recruiting
NCT05489211	II	No	Endometrial Cancer, CRC, mCRPC, Ovarian Cancer, Gastric Cancer	Advanced	Datopotamab Deruxtecan	-	ORR, AE	Recruiting
NCT04484142	II	No	NSCLC With Actionable Genomic Alterations	Advanced	Datopotamab Deruxtecan	NA	ORR	Recruiting
NCT03944772	II	No	NSCLC	Advanced	Datopotamab Deruxtecan with osimertinib	-	ORR	Recruiting
NCT04940325	II	No	NSCLC	Advanced	Datopotamab Deruxtecan	NA	ORR	Recruiting
NCT05104866	III	Yes	HR+, HER2-, Breast cancer	Advanced	Datopotamab Deruxtecan	Eribuline or Capecitabine or Gemcitabine or Vinorelbine	PFS, OS	Recruiting
NCT05374512	III	Yes	Triple-Negative Breast Cancer	Advanced	Datopotamab Deruxtecan	Paclitaxel or Nab-paclitaxel or Carboplatin or Capecitabine or Eribulin	PFS, OS	Recruiting
NCT05215340	III	Yes	NSCLC Without Actionable Genomic Alterations	Advanced	Datopotamab Deruxtecan with Pembrolizumab	Pembrolizumab	PFS, OS	Recruiting
NCT04656652	III	Yes	NSCLC	Advanced	Datopotamab Deruxtecan	Docetaxel	PFS, OS	Recruiting
NCT05555732	III	Yes	NSCLC	Advanced	Datopotamab Deruxtecan with Pembrolizumab With or Without Platinum	Datopotamab Deruxtecan with Pembrolizumab, Platinum and Pemetrexed	PFS, OS	Not yet recruiting

MTD: maximum tolerated dose; RP2D: recommended phase 2 dose; DLT: dose limiting toxicity; AEs: adverse events; SAE: serious adverse events; RDE: recommended Doses for Expansion; ORR: objective response rate; Treatment-emergent Adverse Event (TEAE); iDFS: invasive disease free survival; NSCLC: no small cell lung cancer.

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
