# Peer review of "Overview of Trop-2 in Cancer: From Pre-Clinical Studies to Future Directions in Clinical Settings"

_cancers, 2023, doi:10.3390/cancers15061744_

Round 1
Reviewer 1 Report
This is a timely review article on roles, mechanisms, and potential implications of Trop-2 in cancer from both the preclinical and clinical perspectives. The authors started with Trop-2 biology and its functions, followed with its related pathways in cancer. Its role in cancer treatment and current clinic trials were further introduced with details.
1. The reviewer likes the organization and logic of the manuscript, but feels that the function of Trop-2 and its biological networks are a bit scattered.
2. Figure 1 feels weak to support Section 1. In addition, more figures can be added to supplement contents in the following part.
3. After a quick search on Pubmed, a few studies on Trop-2 targeted molecular imaging have been performed, relevant information can be added to the manuscript.
4. Major mistakes can be found in the manuscript, for example on Line 47-48, “Trop-2 was may similarly confer the capacity for proliferation and invasion to cancer cells.”
5. Acronyms should be spelled out at their first appearance, such as “HIKE”
Author Response
Reviewer #1: This is a timely review article on roles, mechanisms, and potential implications of Trop-2 in cancer from both the preclinical and clinical perspectives. The authors started with Trop-2 biology and its functions, followed with its related pathways in cancer. Its role in cancer treatment and current clinic trials were further introduced with details.
- The reviewer likes the organization and logic of the manuscript, but feels that the function of Trop-2 and its biological networks are a bit scattered.
Thanks for your comment. We tried to implement the biological networks comprehension by adding explanations as necessary without altering the work structure.
- Figure 1 feels weak to support Section 1. In addition, more figures can be added to supplement contents in the following part.
Thank you for this comment. Nevertheless, an extensive review of the biological patterns underlying Trop-2 action exceeds the objective of this work (focuses on clinical behaviour).
- After a quick search on Pubmed, a few studies on Trop-2 targeted molecular imaging have been performed, relevant information can be added to the manuscript.
Thanks. We added some references in the text according to this suggestion.
- Major mistakes can be found in the manuscript, for example on Line 47-48, “Trop-2 was may similarly confer the capacity for proliferation and invasion to cancer cells.”
- Acronyms should be spelled out at their first appearance, such as “HIKE”
#4 and #5 Thank you. We have done an extensive text revision to identify acronyms not spelt, typos and mistakes.
Reviewer 2 Report
The review is interesting but English is at times very difficult to understand and must be corrected. The authors jump from present to past tense so it is hard to understand what was published previously and what is a different paper/study----
Author Response
Reviewer #2: The review is interesting but English is at times very difficult to understand and must be corrected. The authors jump from present to past tense so it is hard to understand what was published previously and what is a different paper/study----
Thank you for your comments. We have done an extensive text revision to identify acronyms not spelt, typos and mistakes.
Round 2
Reviewer 2 Report
I believe the review has been improved sufficiently.